# EMG-JEPA: Towards Scalable and Generalizable sEMG-Based Hand Pose Estimation via Self-Supervised Learning

## Abstract

This work introduces EMG-JEPA, a Joint Embedding Predictive Architecture (JEPA) designed to improve generalization for hand pose estimation from surface electromyography (sEMG) signals. Collecting labeled sEMG data for hand pose estimation is costly, as it requires synchronizing the sEMG recordings with motion capture systems to obtain precise joint-angle annotations. To mitigate the dependency on such expensive labels, EMG-JEPA uses self-supervised learning to derive transferable representations from unlabeled sEMG signals, which can then be fine-tuned for downstream hand pose estimation. We analyze the effectiveness of EMG-JEPA on data collected from three wrist-worn devices, providing signals with 8, 16, and 110 channels. Our results show that EMG-JEPA can improve cross-user hand pose estimation, particularly in high-channel-density settings, reducing joint-angle error by up to 3.55% and 5.13% for the 16- and 110-channel setups, respectively. Further, results from the 8-channel setup suggest a channel-density threshold ($\approx$ 16 channels), below which JEPA-based pretraining offers limited gains. Overall, our study identifies key design choices for developing a JEPA for sEMG, offering a scalable approach to reduce labeled data requirements.

## 1 Introduction

Accurate estimation of hand pose is key to enabling natural and precise human-computer interaction, allowing gestures such as pinching or pointing to map directly to virtual actions, capacities that traditional input devices such as keyboards, mice, and controllers cannot match. Existing approaches, however, have limitations for practical, on-the-go use. Vision-based systems using monocular cameras, depth sensors, or motion capture (Cai et al., 2018; Mueller et al., 2017; Ge et al., 2016; Han et al., 2018; Yuan et al., 2018; Park et al., 2020) can capture detailed hand movements, but they are constrained by occlusions, lighting conditions, and restricted fields of view. Glove-based wearables (Luo et al., 2021; Yang et al., 2021; Truong et al., 2018; Shen et al., 2016; Tashakori et al., 2024) provide precise finger-joint measurements, but they can impede dexterous manipulation and interfere with natural motion. Forearm-mounted devices (Zhang & Harrison, 2015; Laput & Harrison, 2019; Laput et al., 2016; McIntosh et al., 2017), while more comfortable, are at a disadvantage for collecting data at scale because they are more encumbering for users. Surface electromyography (sEMG) from wrist-worn sensors offers a compelling alternative: it captures rich, high-dimensional muscle signals directly linked to finger and hand movements, enables continuous recording at scale without restricting motion, and avoids the occlusion and line-of-sight limitations inherent to vision-based methods.

Recent efforts have explored mapping sEMG signals directly to hand pose using supervised learning, where models are trained on synchronized sEMG and motion capture data (Salter et al., 2024; Quivira et al., 2018; Sosin et al., 2018; Liu et al., 2021; Sîmpetru et al., 2022). These approaches have demonstrated promising accuracy within individual users, but they tend to generalize less effectively across users due to the high variability of sEMG signals caused by differences in muscle physiology, electrode placement, and skin impedance (Sussillo et al., 2024; Liu et al., 2021). Moreover, acquiring large-scale labeled datasets is labor-intensive and costly, as it requires precise temporal alignment between muscle activity and ground-truth joint angles.

To overcome these challenges, we introduce EMG-JEPA, a self-supervised model based on the Joint-Embedding Predictive Architecture (JEPA) (LeCun, 2022). Self-supervised learning enables the extraction of meaningful representations from unlabeled data by training models to predict relationships within the data itself, rather than relying on external annotations. This is particularly valuable for sEMG, where collecting unlabeled data is far easier and more scalable across users, as it does not require sophisticated setups such as infrared motion capture or optical tracking to obtain ground-truth hand trajectories. The JEPA framework has recently demonstrated strong performance across diverse modalities (images (Assran et al., 2023), videos (Bardes et al., 2024a; Assran et al., 2025), and brain signals (Dong et al., 2024)) by learning context-aware representations that capture invariant structure without reconstructing raw input. Building on this foundation, we develop EMG-JEPA, which leverages temporal context and cross-channel dependencies in sEMG signals to learn representations that can be fine-tuned for downstream hand pose estimation.

We evaluate EMG-JEPA on sEMG datasets collected from three wrist-worn devices that provide signals with 8, 16, and 110 channels, allowing us to systematically study how input dimensionality influences representation learning. Our experiments show that EMG-JEPA improves cross-user generalization over supervised baselines, reducing joint-angle error by 3.55% and 5.13% for the 16- and 110-channel setups, respectively. Interestingly, these benefits disappear in the 8-channel configuration, revealing that effective representation learning with JEPA requires sufficient cross-channel correlation. Together, these results demonstrate that EMG-JEPA enhances cross-user generalization and mitigates dependence on extensive labeled datasets, enabling more scalable sEMG-based interaction systems.

Our main contributions can be summarized as follows:

- We introduce EMG-JEPA, a self-supervised learning framework based on the Joint-Embedding Predictive Architecture (JEPA) for learning robust, transferable representations from unlabeled surface electromyography (sEMG) signals. We highlight key design choices and modeling strategies needed to capture meaningful signal structure, including input signal representation, masking strategies for JEPA training, and the use of temporal and channel embeddings.

- We present a comprehensive cross-user evaluation across three sensor configurations (8, 16, and 110 channels) and show that EMG-JEPA improves sEMG-based hand pose estimation. Notably, we evaluate EMG-JEPA for direct sEMG-to-pose modeling, without intermediate velocity estimation or user-specific tuning, thereby simplifying deployment and improving generalization.

- Through our systematic cross-sensor evaluation, we uncover a critical dimensionality threshold: effective sEMG representation learning via masked latent prediction requires sufficient inter-channel correlation to capture coherent muscle activation patterns.

## 2 Related Work

**sEMG-Based Hand Pose Estimation.** Recent works in sEMG-based hand pose estimation indicate that cross-user generalization benefits from large labeled datasets. For example, Salter et al. (2024) introduce the emg2pose benchmark with 193 participants and 370 hours of synchronized 16-channel sEMG and motion capture data, demonstrating power-law scaling that motivates large-scale pretraining. Complementing this, Sussillo et al. (2024) demonstrate that wrist-worn sEMG systems can generalize to 6,627 users in a zero-shot setting, with additional gains achievable through personalization. However, collecting such labeled data is labor-intensive, expensive, and difficult to scale to more users or diverse conditions. Approaches that account for anatomical variability, such as anatomy-aware geometric learning for high-density sEMG (Dash et al., 2025), improve cross-user performance, while studies on scaling and sensor placement (Botros et al., 2025; Eddy et al., 2024) demonstrate the influence of channel density and dataset size. Multimodal extensions that combine sEMG with IMU or video (Xiao et al., 2025; Yin et al., 2025) further highlight the benefits of richer signal representations. Collectively, these works motivate self-supervised approaches capable of leveraging unlabeled sEMG data to learn scalable and generalizable representations.

**Self-Supervised Learning for Biosignals.** Self-supervised learning (SSL) has become critical for biosignal analysis due to the high cost of labeling physiological data. For sEMG, Raghu et al. (2025) apply VICReg contrastive learning to LSTM-based models trained on unlabeled continuous activations, showing that unsegmented gesture transitions and time-shift augmentations improve downstream performance. Abbaspourazad et al. (2024) train contrastive SSL models on ECG and PPG from 141,000 Apple Watch users, revealing embeddings that encode demographic and health features for downstream transfer. To improve cross-subject robustness, Cheng et al. (2020) propose a subject-aware contrastive framework for EEG and ECG that enforces invariance through adversarial loss. Multimodal progress includes PhysioWave (zha, 2025), a wavelet-transformer architecture for EEG, ECG, and sEMG that captures frequency and temporal structure, and BioCodec (Avramidis et al., 2025), a tokenization method using discrete codebooks to handle low signal-to-noise physiological data. Beyond contrastive approaches, masked prediction paradigms have proven effective for multi-channel signals. For example, Fu et al. (2024) introduce a graph-based masked autoencoder using anatomical priors, Chen et al. (2024) reconstruct multi-lead ECG via channel masking, and Muna et al. (2025) present a dual-path model for EEG capturing both local and global dependencies. Together, these advances show that SSL can learn robust biosignal representations, though most focus on classification rather than high-dimensional regression tasks like continuous pose estimation.

**Joint Embedding Predictive Architectures.** Joint Embedding Predictive Architecture (JEPA) (Le-Cun, 2022) learns by predicting latent representations of masked regions from visible context, avoiding reconstruction or contrastive negatives, and achieves strong efficiency and performance for computer vision applications (Assran et al., 2023; Bardes et al., 2024a; Assran et al., 2025). Extensions to other modalities include audio (Fei et al., 2024; Tuncay et al., 2025), language (Huang et al., 2025), 3D point clouds (Saito et al., 2025), and optical flow (Bardes et al., 2024b). Recent works have even extended this architecture to multivariate time series data (Dutta et al., 2025; Ennadir et al., 2025) and brain signals Dong et al. (2024). These studies demonstrate that spatiotemporal and semantic masking can enable efficient representation learning across modalities, motivating our architectural choices. Yet, despite JEPA's success, its application to multi-channel physiological signals like sEMG remains limited.

**Learning from High-Density Multichannel Signals.** Recent work shows that moving from low-channel to high-channel sEMG increases spatial resolution, which, when paired with architecture-aware designs, can substantially improve sEMG-based modeling (Montazerin et al., 2023; Shin et al., 2024; Brüsch et al., 2024; Mehlman et al., 2025). These works highlight the trade-off between model complexity and channel density, motivating our investigation of the minimum electrode resolution needed for effective JEPA-based masked prediction and practical wearable design.

Building on advances in sEMG modeling, self-supervised learning for biosegnals, and joint embedding predictive architectures, EMG-JEPA introduces one of the first latent-predictive frameworks for cross-user hand pose estimation from raw multi-channel sEMG. This enables robust representation learning without labeled data or user-specific calibration.

## 3 Method

In this section, we present our approach to hand pose estimation from sEMG signals, leveraging self-supervised learning (SSL) followed by supervised fine-tuning (SFT). We present the EMG-JEPA, an adaptation of the Joint Embedding Predictive Architecture (JEPA) for multi-channel time-series sEMG, which learns to predict latent representations of masked signal segments from unlabeled data for robust feature extraction. SFT with a lightweight regression head then enables accurate joint angle prediction.

Section 3.1 outlines the signal processing and representation strategies used to prepare raw sEMG data for the model. Section 3.2 details the EMG-JEPA framework, and Section 3.3 describes the fine-tuning procedure for hand pose estimation.

### 3.1 Signal Processing and Representation

Figure 1 provides a high-level overview of the process of converting raw sEMG signals into input tokens for the model.

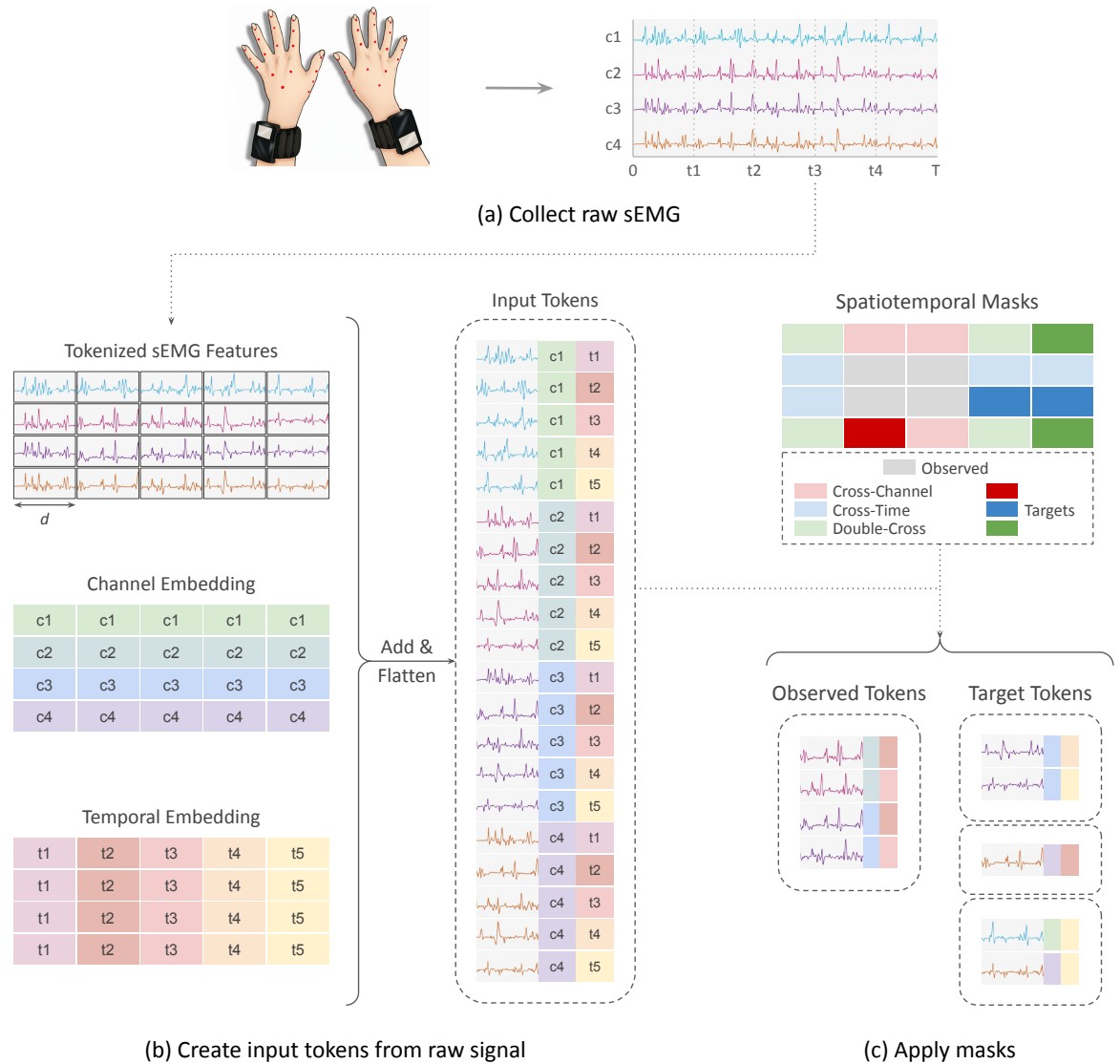

Figure 1: Preparation of sEMG signals for EMG-JEPA. (a) Raw sEMG signals are collected using wrist-worn sensor bands (e.g, sEMG-RD wristband (Sussillo et al., 2024)), which provides $C$ channels (only 4 shown for illustration). (b) The signals are tokenized, projected into $d$-dimensional feature space, and added with channel and temporal embeddings. (c) Spatiotemporal masking is applied to partition the sequence into observed and target tokens. Best viewed in color.

### 3.1.1 Preprocessing

The raw sEMG signals, sampled at 2 kHz, are first bandpass filtered between 20 and 850 Hz to remove low-frequency drift and high-frequency noise (Salter et al., 2024). Next, we perform channel-wise median subtraction to center each channel, followed by channel-wise L2 normalization to scale the signals. Both the median and L2 norm are computed independently for each channel and for each user, ensuring that the preprocessing accounts for inter-user variability and preserves relative signal patterns within each channel. The target hand joint angles (sampled at 60 Hz) are smoothed with a 15 Hz low-pass filter to eliminate jitter.

### 3.1.2 Tokenization and Featurization

To feed the preprocessed sEMG signals into our model, we convert them into tokens. Given a signal $X \in \mathbb{R}^{C \times T}$ with $C$ channels and $T$ time steps, each channel is split into non-overlapping patches of size $t$, producing $K = \lfloor T/t \rfloor$ tokens. This results in a tokenized representation $X_{\text{tok}} \in \mathbb{R}^{C \times K \times t}$.

The tokenized input is passed through a multi-layer perceptron that projects the raw signal tokens into the model's feature space. This converts $X_{\text{tok}} \in \mathbb{R}^{C \times K \times t}$ into featurized representations $X_{\text{feat}} \in \mathbb{R}^{C \times K \times d}$, where $d$ is the model's feature dimension. This projection is learned jointly with the rest of the network during training.

Prior works using Transformer-based models for sEMG or other 1D signals have favored convolution-based featurization of the input signal (Mehlman et al., 2025; Schneider et al., 2019). However, using convolutions in our setup would compromise the integrity of masked prediction, as information from masked tokens could leak into the representation of their unmasked neighbors within the receptive field of convolutional layers. By using independent featurization via an MLP, we ensure that each token's representation is derived solely from its corresponding input signal, preserving the validity of the masking strategy and enabling robust learning.

### 3.1.3 Temporal and Channel Embeddings.

To encode temporal and channel information for each token, we use two types of embeddings:

1. Temporal embeddings $E_{\text{temp}} \in \mathbb{R}^{K \times d}$: Sinusoidal embeddings that encode the temporal position of each of the $K$ patches. Patches across all channels that correspond to the same time step share the same temporal embedding.

2. Channel embeddings $E_{\text{chan}} \in \mathbb{R}^{C \times d}$: Sinusoidal embedding that encodes the channel identity. All patches that pertain to the same channel share the same channel embedding.

These embeddings are added to the featurized input tokens to provide both temporal and channel-specific positional information to the model:

$$X_{\text{emb}} = X_{\text{feat}} + E_{\text{temp}}^{[\mathbf{1}_C, :, :]} + E_{\text{chan}}^{[:, \mathbf{1}_K, :]},$$

where $E_{\text{temp}}^{[\mathbf{1}_C, :, :]}$ denotes the temporal embedding broadcasted along the channel dimension and $E_{\text{chan}}^{[:, \mathbf{1}_K, :]}$ denotes the channel embedding broadcasted along the temporal dimension. Finally, $X_{\text{emb}}$ is flattened along the temporal and channel dimensions to give $X_{\text{inp}} \in \mathbb{R}^{N \times d}$, where $N = C \cdot K$. This produces a sequence of $N$ tokens, each $d$-dimensional, which serves as the input to our Transformer-based model described next.

## 3.2 EMG-JEPA

EMG-JEPA adapts the Joint Embedding Predictive Architecture (JEPA) to represent multi-channel sEMG. Raw sEMG signals are converted into a sequence of tokens, a masking strategy hides segments of the sequence, and the model learns to predict latent representations of the missing tokens from the visible context. This guides the network to capture temporal dependencies and cross-channel relationships within the input signal without external supervision.

### 3.2.1 Spatiotemporal Masking

We apply spatiotemporal masks to extract the observed context (the visible portion of the signal) and the target (the portion to be predicted) from the input. The observation block $X_{\text{obs}}$ is formed by randomly selecting a contiguous segment in both the temporal and channel dimensions, and the target block $X_{\text{tgt}}$ is sampled from the remaining regions of $X_{\text{inp}}$. Following Dong et al. (2024), we define three target categories: (1) Cross-channel, requiring the prediction of unseen channels at observed timesteps; (2) Cross-time, requiring the prediction of unseen timesteps within observed channels; and (3) Double-cross, requiring the prediction

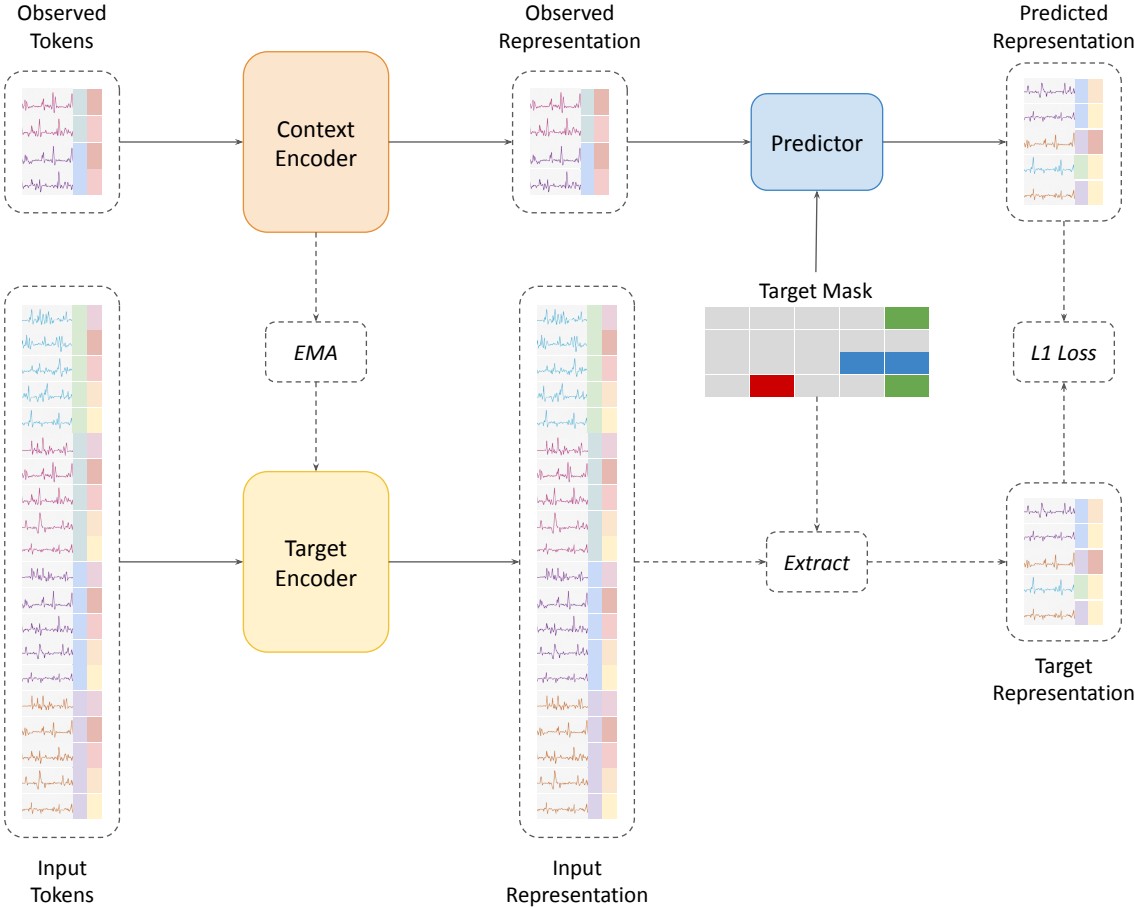

Figure 2: Overview of EMG-JEPA. The context encoder maps the observed tokens into a latent representation. The target encoder maps the full input into latent representations, from which target representations are extracted. The predictor takes representations of the observed tokens and the target block mask as input, and generates predictions for the masked target regions. $L1$ loss is applied between the predicted and target representations. The context encoder and predictor are learned using gradient optimization, and the target encoder is updated via exponential moving average of the context encoder.

of unseen channels at unseen timesteps. During training, a target block is randomly sampled from each category.

Given the circular arrangement of sEMG electrodes around the wrist, masks can wrap around the channel dimension to ensure continuity. For instance, a valid mask for a 16-channel sEMG signal can include a temporally contiguous segment across channels 1, 2, 15, and 16. The block size is determined by the range ratios $\{\eta^C, \eta^T\}$, where $\eta^C$ specifies the fraction of channels and $\eta^T$ specifies the fraction of time patches to include. We experimentally find that setting $\eta^C, \eta^T \in [0.4, 0.6]$ for the observed block, and $\eta^C, \eta^T \in [0.2, 0.4]$ for the target blocks, leads to stable training, especially for our 8- and 16-channel setups. This differs from other JEPA models (Assran et al., 2023; Bardes et al., 2024a; Assran et al., 2025), which use smaller observed blocks and larger target blocks. We conjecture that the inter-channel correlations are relatively weak with 8 and 16 channels, due to which using too small an observed block would provide insufficient context for the model to accurately predict the masked targets. The 110-channel setup is more robust to a larger target block, however, for consistency, we use the same size ratios for all three setups.

### 3.2.2 Architecture

EMG-JEPA consists of three main components: (1) Context Encoder, which maps the observed context $X_{\mathrm{obs}}$ into a latent representation $Z_{\mathrm{obs}}$, capturing temporal and cross-channel dependencies; (2) Target Encoder, which maps the input $X_{\mathrm{inp}}$ into a latent representation $Z_{\mathrm{inp}}$, from which the representations of the target blocks $Z_{\mathrm{tgt}}$ are extracted; and (3) Predictor, which takes the representation of the observed context $Z_{\mathrm{obs}}$ as input and, conditioned on the target block mask, generates predictions for the corresponding target representations $\hat{Z}_{\mathrm{tgt}}$.

All three components are implemented as Transformer models (Vaswani et al., 2017). The context and target encoders share the same architecture, each consisting of 6 Transformer blocks with a feature dimension of 768. The predictor is a lighter model with 4 Transformer blocks and a reduced feature dimension of 384. It includes two linear projections: an input projection that maps the 768-dimensional encoder output to 384, and an output projection that maps the 384-dimensional predictor output back to 768.

### 3.2.3 Training

The model is trained to minimize the $L_1$ distance between the target encoder's representations of the target block $Z_{\mathrm{tgt}}$ and the predictor's output $\hat{Z}_{\mathrm{tgt}}$. The parameters of the context encoder and predictor are learned through gradient-based optimization, while the parameters of the target encoder are updated using the exponential moving average of the context-encoder parameters. More details on the training configuration and hyperparameters are provided in Appendix A.1.

By predicting the masked part of the sEMG signal, the model learns to capture meaningful patterns of muscle activation rather than relying on trivial local correlations. By including cross-channel, cross-time, and double-cross targets, the model learns to generalize to unseen channels and unseen time segments, which mimics real-world scenarios such as variations in electrode placement or user-specific movement patterns. Moreover, predicting in latent space prevents overfitting to raw signal values and encourages the model to extract higher-level features that are more consistent across users and gestures.

### 3.3 Fine-Tuning for Hand Pose Estimation

After self-supervised pretraining, we fine-tune EMG-JEPA for hand pose estimation by attaching a lightweight regression head on top of the pretrained context encoder. The predictor and target encoder are discarded, as they are only required during pretraining. We note that the target encoder could alternatively be used in place of the context encoder for fine-tuning; however, we observe no difference in performance between the two choices.

Given the encoder output $Z \in \mathbb{R}^{N \times d}$, the regression head applies attention-based pooling: a learned query attends over the $N$ tokens to produce a single pooled representation. This pooled token is then passed through a two-layer MLP with ReLU activation to generate the final joint angle predictions $Y \in \mathbb{R}^J$, where $J = 22$ corresponds to the number of hand and finger joint angles being predicted (i.e., the degrees of freedom of the hand model).

During fine-tuning, all parameters of the context encoder and regression head are updated jointly. We use a mean squared error (MSE) loss between the predicted and ground-truth joint angles as the optimization objective. To prevent overfitting, we apply early stopping based on validation performance and employ dropout within the regression head. More details on the fine-tuning configuration and hyperparameters are provided in Appendix A.2.

This fine-tuning stage enables the model to adapt the general-purpose representations learned during pretraining to the specific task of hand pose estimation. As shown in Section 4.2, this approach holds the potential to improve generalization across participants compared to training purely supervised models from scratch.

## 4 Experiments

In this section, we discuss the sEMG datasets used in our study (Section 4.1), the cross-user generalization results (Section 4.2), and the ablation experiments for our design choices (Section 4.3). We also investigated strategies to improve performance in the 8-channel setup (Section 4.4); while these approaches did not yield significant gains, they motivated our conjecture regarding the dimensionality threshold for effective representation learning.

### 4.1 Data

We evaluate EMG-JEPA using sEMG recordings collected from three wrist-worn sensor configurations, providing 8-, 16-, and 110-channel signals, each sampled at 2 kHz. The datasets were designed to capture diverse hand and finger movements, including gestures such as pinching, pointing, and grasping, as well as continuous hand motions, ensuring a broad coverage of naturalistic articulation. For the 8- and 16-channel EMG signals, ground-truth joint angle annotations are obtained from head-mounted display images using a vision-based hand tracking system, UmeTrack (Han et al., 2022). For the 110-channel EMG signal, ground-truth joint angles are obtained using a motion capture system. In all cases, the joint angles were sampled at 60 Hz.

Each recording session involved participants performing multiple repetitions of prompted movements. Sessions were structured to capture intra-subject variability, with the majority of participants completing multiple sessions per device placement. During each session, participants were guided by visual cues and verbal instructions to move their hands through a wide range of postures and trajectories, including motions across the body and between waist and shoulder levels, ensuring varied hand configurations. All participants provided informed consent prior to data collection, and the study followed institutional ethical guidelines. No personally identifiable information was collected or used at any stage of the data recording or analysis.

For training, we use 181, 684, and 305 sessions for the 8-, 16-, and 110-channel setups, respectively. An additional 30 sessions were reserved for testing across all configurations to evaluate cross-user generalization. Data splits were defined at the participant level, ensuring that no test participant was seen during training.

### 4.2 Cross-User Generalization Results

We evaluate EMG-JEPA on the task of cross-user hand pose estimation, where the model is trained on sEMG recordings from a subset of participants and tested on held-out users. The model takes as input a 1-second window of sEMG recordings and predicts the hand pose corresponding to the final timestamp of that window. Training and evaluation are conducted in a cross-user setting, where the model is trained on recordings from a subset of participants and tested on held-out users to assess generalization. Performance is evaluated using the Joint Angle Mean Absolute Error (JA-MAE), which is defined as

$$\text{JA-MAE} = \frac{1}{J} \sum_{j=1}^{J} \left| Y_j - \hat{Y}_j \right|,$$

where $Y_j$ and $\hat{Y}_j$ are the ground truth and predicted angles (in degrees) for joint $j$. Lower MAE indicates better pose estimation accuracy.

To ensure a fair comparison, we define a fully supervised baseline that uses the same architecture as the EMG-JEPA setup (consisting of the context encoder followed by the cross-attention-based decoder), but it is trained end-to-end in a fully supervised manner without any JEPA pretraining. This ensures that both models have identical model capacity and structure, allowing us to isolate the benefits of self-supervised pretraining on cross-user generalization.

Table 1 summarizes the cross-user JA-MAE for EMG-JEPA and the fully supervised baseline across the three channel configurations. EMG-JEPA improves cross-user generalization over the supervised baseline in the 16- and 110-channel setups, reducing JA-MAE by 3.55% (from 15.21° to 14.67°) for the 16-channel setup and 5.13% (from 8.37° to 7.94°) for the 110-channel setup. These results demonstrate that EMG-JEPA's

Table 1: Cross-user hand pose estimation results for EMG-JEPA compared to a fully supervised baseline. Lower is better.

| Method | 8-channel JA-MAE (°) | 16-channel JA-MAE (°) | 110-channel JA-MAE (°) |
|---|---|---|---|
| Fully Supervised | 15.91 | 15.21 | 8.37 |
| EMG-JEPA + Finetuning | 15.90 | **14.67** | **7.94** |

Table 2: Ablation of input normalization. Normalizing each channel per user substantially reduces inter-user variability and improves pose estimation accuracy.

| Normalization | JA-MAE (°) |
|---|---|
| No normalization | 9.42 |
| Median + L2 normalization | 7.94 |

Table 3: Ablation of input featurization. MLP-based featurization preserves independence between tokens and avoids leakage during masked prediction.

| Featurization | JA-MAE (°) |
|---|---|
| Raw signal (no featurization) | 10.05 |
| Convolutional featurization | 8.10 |
| MLP featurization | 7.94 |

self-supervised pretraining is particularly beneficial for high-channel-density sEMG signals. In contrast, for the 8-channel setup, EMG-JEPA provides no improvement over the baseline (15.9° for both), highlighting a dimensionality threshold: low-channel inputs provide insufficient cross-channel information for effective JEPA pretraining. We discuss this in more detail in Section 4.4.

### 4.3 Ablation Experiments

We perform ablation studies on input preprocessing, masking strategy, and model size. Unless otherwise noted, all ablations are conducted on the 110-channel setup.

#### 4.3.1 Input Preprocessing

**Normalization.** We compare performance with and without normalization, where normalization includes median-centering and L2-scaling of each channel, as described in Section 3. Table 2 shows that normalization substantially improves performance, reducing joint-angle MAE from 9.42° (without normalization) to 7.94° (with normalization). This result highlights that scaling and centering the sEMG signals is essential for mitigating inter-subject variability, as raw sEMG amplitudes can vary significantly across users and sessions.

**Signal Featurization.** We also study the effect of different input featurization strategies in Table 3. Feeding the raw signal values directly to the Transformer yields a joint-angle MAE of 10.05°, showing that the model struggles to learn robust representations from unprocessed inputs. Using convolutional featurization improves performance to 8.10°, as the convolution aggregates local temporal patterns. However, this approach introduces information leakage during masked prediction (see Section 3.1.2), undermining the self-supervised training objective. In contrast, MLP-based featurization, which processes each token independently, achieves the best performance (7.94°), preserving the integrity of the masked prediction task and providing a robust representation that generalizes across users.

Together, these ablations demonstrate that both normalization and careful featurization are critical for EMG-JEPA to learn high-quality, generalizable sEMG representations. Normalization helps handle the inter-user variability, while MLP-based token featurization ensures robust self-supervised learning without data leakage, forming a key part of the model's design.

#### 4.3.2 Masking Strategy

We compare different masking strategies to understand their impact on representation learning for sEMG-based pose estimation, and the results are summarized in Table 4.

Table 4: Ablation of masking strategy. Structured spatiotemporal masking using cross-channel, cross-time, and double-cross strategy (Section 3.2.1) performs better than random masking strategies.

| Masking Strategy | JA-MAE (°) |
|---|---|
| Random token | 8.38 |
| Random block | 8.01 |
| Cross-block (ours) | 7.94 |

Table 5: Ablation of decoder architecture. JEPA pretraining consistently outperforms randomly initialized encoders across both convolutional and cross-attention decoders.

| Decoder Type | Encoder Initialization | JA-MAE (°) |
|---|---|---|
| Convolutional | Random | 8.41 |
| Convolutional | JEPA | 8.12 |
| Cross-attention | Random | 8.37 |
| Cross-attention | JEPA | 7.94 |

The random token masking strategy randomly selects individual tokens across the input sequence for both observed and target sets. This approach achieves similar performance to the fully supervised baseline, as the model primarily learns local interpolation during self-supervised JEPA pretraining, rather than meaningful contextual representations.

The random block masking strategy randomly samples contiguous spatiotemporal regions as the observed context and targets. This provides stronger contextual learning than token-level masking, as the model must infer missing information from broader temporal and spatial cues.

Finally, the proposed cross-block masking (Section 3.2.1) selects target regions from cross-channel, cross-time, and double-cross locations (Figure 1(c)), enforcing reasoning across non-local dependencies. This structured masking achieves the best performance, indicating that it better leverages the spatiotemporal structure of sEMG signals.

### 4.3.3 Decoder Architecture

We evaluate two decoder architectures for predicting hand pose from the encoder's latent representations: (1) a convolutional decoder, which applies a series of 1D convolutions over the temporal dimension of the encoder outputs to regress joint angles; and (2) a cross-attention decoder, which uses a learnable query token to attend to all latent representations through a cross-attention block followed by an MLP (Section 3.3). As shown in Table 5, JEPA pretraining improves performance in both cases compared to the fully supervised approach, indicating that the benefits of self-supervised pretraining generalize across decoder designs.

### 4.4 Analysis of Low-Channel Configurations

To investigate the limited gains in the 8-channel setup, we applied standard data augmentation (e.g., Gaussian noise, temporal jitter) and interpolated the signals to higher-dimensional inputs (16- and 110-channel), but these approaches did not yield any measurable improvement. We then explored two more sophisticated strategies specifically designed to enhance EMG-JEPA's performance under low-channel constraints.

**Emulated 8-channel signals from high-density recordings.** We examined whether augmenting training data could improve EMG-JEPA's performance by emulating 8-channel signals from the 110-channel dataset. This emulation was achieved by projecting the spatial layout of the 8 electrodes onto the 110-channel grid and interpolating the corresponding channels to synthesize equivalent 8-channel signals. These emulated signals were used in three different ways: (1) only for JEPA pretraining, (2) only for pose decoder training, and (3) for both JEPA pretraining and pose decoder training. Among these, using the emulated data solely for JEPA pretraining gave marginally better performance, but none of the approaches produced any significant improvement over the fully supervised baseline. This suggests that the limitation is not the amount of training data but the representational capacity of low-channel inputs.

**Cross-configuration adaptation via learned projection.** We trained an adapter network to project the 8-channel signals into a pseudo–110-channel representation. This approach was motivated by the hypothesis that projecting low-channel inputs into a higher-dimensional 110-channel space could allow the use of a

Table 6: Cross-user hand pose estimation for the 8-channel setup. Neither emulated 8-channel signals nor cross-configuration adaptation via a learned adapter improves over the fully supervised baseline.

| Method | JA-MAE (°) |
|---|---|
| Fully Supervised | 15.91 |
| Emulating signal (110-channel → 8-channel) | 15.84 |
| Adapter network (8-channel → 110-channel) | 15.95 |

context encoder pretrained on the richer 110-channel signals, which have been shown to improve cross-user generalization. The pose decoding pipeline in this setup consists of an MLP adapter that maps the 8-channel input to 110 channels, followed by the pretrained 110 channel context encoder and the pose decoder trained for the 110 channel setup. The training was performed in three stages:

1. Adapter-only training: The MLP adapter is trained while keeping both the context encoder and pose decoder frozen.

2. Adapter + decoder training: The pose decoder is unfrozen and trained jointly with the adapter, while the context encoder remains frozen.

3. Full fine-tuning: The context encoder is also unfrozen and trained jointly with the adapter and decoder.

Smaller learning rates were used for the pretrained components to prevent catastrophic forgetting of representations learned for 110-channel setup (pose decoder: $1 \times 10^{-4}$, context encoder: $1 \times 10^{-5}$), while the adapter was trained with a higher learning rate ($5 \times 10^{-4}$) to facilitate learning from the sparse input. Despite this staged training strategy, the adapter network did not yield measurable improvements over the fully supervised baseline, indicating that the missing spatial information in the 8-channel signals cannot be recovered through learned projection alone.

**Conclusions on Sparse EMG Inputs.** Together, these findings reinforce our conclusion that EMG-JEPA's effectiveness hinges on sufficient cross-channel diversity in the input signal. Since JEPA learns by predicting masked representations conditioned on their unmasked neighbors, it relies on meaningful correlations across channels to infer missing information. In the 8-channel configuration, the spatial sampling of muscle activity is too sparse for such correlations to emerge, making the predictive objective ill-posed. In contrast, configurations with 16 or more channels provide denser spatial coverage of wrist, enabling the JEPA objective to leverage inter-channel structure—thereby facilitating effective representation learning and improved cross-user generalization.

## 5 Conclusion

This work introduced EMG-JEPA, a self-supervised model for learning transferable representations from surface electromyography (sEMG) signals for cross-user hand pose estimation. By adapting the Joint-Embedding Predictive Architecture (JEPA) framework to the sEMG domain, EMG-JEPA enables predictive representation learning without relying too heavily on costly motion-capture-based supervision. Through extensive experiments across three sensor configurations, we demonstrated that EMG-JEPA consistently improves generalization in higher-channel setups while revealing a dimensionality threshold below which the predictive learning objective becomes ineffective. Our analyses highlight key design insights for self-supervised sEMG modeling for robust representation learning.

## 6 Limitations and Future Work

While EMG-JEPA demonstrates strong performance in cross-user hand pose estimation, several limitations remain, which also serve as inspirations for future research.

**Model Size and Architecture.** Our transformer-based JEPA and pose decoder are larger than most existing approaches, which often use compact architectures such as time–depth separable convolution. This increased model size may limit on-device deployment, particularly for wearable or real-time applications. A recent study has shown that large transformer models for EMG can be distilled into networks up to $50\times$ smaller with minimal loss in performance (Mehlman et al., 2025), suggesting a promising direction for future work, though we do not explore such compression techniques in this study. Besides Mehlman et al. (2025), there have been limited explorations on using Transformer-style architectures for EMG signals, as most approaches still rely on convolutional or LSTM-based models. Investigating more expressive attention mechanisms, cross-channel interactions, or hierarchical temporal modeling could further improve representation quality and generalization across users and devices.

**Pretraining cost.** Although self-supervised pretraining reduces reliance on labeled data, it introduces additional computational costs compared to purely supervised approaches. However, this cost can be considered a one-time investment that may accelerate user-specific personalization. We do not explore this aspect in this work, but future studies could investigate how pretraining facilitates rapid adaptation to new users. Future work could also investigate more efficient training strategies or lightweight model variants to make pretraining more accessible for resource-constrained settings.

**Task scope.** Our evaluation is limited to continuous hand pose estimation. While this serves as a challenging setup for assessing representation quality, extending EMG-JEPA to related domains, such as force or stiffness prediction, activity recognition, or multimodal fusion with IMU or video, would provide a broader assessment of its generality and utility.

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

# A    Training Details

## A.1    EMG-JEPA Training

EMG-JEPA is trained to predict the target encoder's latent representations of masked signal regions from the context encoder's output, using an $L_1$ reconstruction loss between the predicted and target embeddings. The context encoder and predictor are optimized jointly, while the target encoder parameters are updated as an exponential moving average (EMA) of the context encoder with a decay factor of 0.996.

A lightweight MLP-based featurizer is applied to each patch independently, projecting it from the raw temporal domain into a 384-dimensional feature space through an intermediate layer of 256 units with GELU activation. This design preserves independence between patches and prevents data leakage during masked prediction.

The featurized patches are then processed by a Transformer encoder with 6 layers, 8 attention heads, a hidden dimension of 768, an MLP dimension of 1536, and a dropout rate of 0.1. The predictor network mirrors this architecture, but it uses a smaller hidden dimension of 384.

For masking, the spatiotemporal masking strategy randomly samples target blocks covering 20–40% of the temporal span and 20–40% of the channels, while observation blocks cover 40–60% along both dimensions.

Training is performed for 10 epochs using the AdamW optimizer with a learning rate of $10^{-4}$, $\beta_1 = 0.9$, $\beta_2 = 0.999$, gradient clipping at 5.0, and linear warmup over 1000 steps. All experiments are conducted on 16 NVIDIA A100 GPUs (2 nodes $\times$ 8 GPUs per node).

## A.2    Pose Decoder Training

For pose estimation, we train a decoder network that maps the encoder's latent representations to joint-angle predictions. The encoder architecture and preprocessing steps follow those described in Appendix A.1 (JEPA Training). When initializing from the pretrained EMG-JEPA checkpoint, we use the context encoder as the encoder for pose decoding. Each training sample corresponds to a 1-second window of sEMG data sampled at 2000 Hz, paired with the ground-truth hand pose recorded at the end of the window.

The decoder adopts a cross-attention architecture, where a learnable query token attends to all latent representations from the encoder through a cross-attention block, followed by a lightweight MLP head for regression. The decoder uses a hidden dimension of 256, MLP dimension of 1536, 8 attention heads, and a dropout rate of 0.1, followed by a linear projection to 20 output dimensions corresponding to the hand joint angles.

Training is performed for up to 25 epochs using the AdamW optimizer with a learning rate of $10^{-4}$, $\beta_1 = 0.9$, $\beta_2 = 0.999$, gradient clipping at 5.0, and linear warmup over 1000 steps. We apply early stopping based on validation JA-MAE to prevent overfitting. All experiments are conducted on 16 NVIDIA A100 GPUs (2 nodes $\times$ 8 GPUs per node).

# B    Broader Impact

The introduction of EMG-JEPA represents an important step toward scalable and generalizable modeling of muscle activity from surface electromyography (sEMG). By enabling self-supervised representation learning for cross-user hand pose estimation, EMG-JEPA bridges the gap between efficient biosignal modeling and practical deployment in human–machine interaction systems. Its broader impact extends across healthcare, assistive technologies, and the study of motor control.

**Biomedical and Assistive Applications.** EMG-JEPA's ability to extract robust, user-invariant representations from unlabeled sEMG signals can advance prosthetic control, rehabilitation robotics, and neuromuscular diagnostics. Improved cross-user generalization reduces the need for subject-specific calibration, making prosthetic and assistive systems more accessible and adaptive to new users—especially individuals for whom collecting extensive calibration data is impractical, such as those with physical impairments.

Moreover, by learning fine-grained muscle activation patterns without labeled data, EMG-JEPA can facilitate early detection of motor impairments or muscular disorders, supporting personalized and continuous monitoring in clinical settings.

**Technological Impact.** By adapting the Joint Embedding Predictive Architecture to multi-channel physiological signals, EMG-JEPA opens a pathway for unified self-supervised modeling across biosignal modalities such as EMG, EEG, and ECG. This approach encourages the development of scalable, label-efficient models that generalize across users, devices, and sensing configurations. The framework's focus on spatiotemporal representation learning may also inspire next-generation wearable systems capable of continuous, privacy-preserving monitoring and intuitive human–computer interfaces.

**Ethical and Social Considerations.** As with any model trained on physiological data, ethical deployment of EMG-JEPA requires careful attention to data privacy, user consent, and demographic fairness. While self-supervised pretraining reduces the dependence on large labeled datasets, ensuring that training data are diverse and representative remains crucial to avoid biases that could impact accessibility or performance across populations. In assistive and medical contexts, transparent reporting and equitable access are essential to ensure that advances enabled by EMG-JEPA benefit all users, not only those with access to high-quality sensing hardware or clinical infrastructure.

