# OpenReview forum: "EMG-JEPA: Towards Scalable and Generalizable sEMG-Based Hand Pose Estimation via Self-Supervised Learning"
_TMLR — Withdrawn by Authors_

### Review · Reviewer_ZA17 · 2026-02-22

**Summary Of Contributions:**

### **Overview**
This paper presents EMG-JEPA, a method applying the Joint Embedding Predictive Architecture (JEPA) to hand pose estimation using surface electromyography (sEMG) signals. The design of EMG-JEPA incorporates time and channel embeddings along with spatiotemporal masks. These components are integrated to learn contextual information across both temporal and channel dimensions of sEMG data. The model's architecture is structured according to the JEPA framework, consisting of a context encoder, a target encoder, and a predictor. The performance of EMG-JEPA was assessed across three sEMG datasets, which featured varying channel densities (8, 16, and 110 channels). The experimental results indicate that EMG-JEPA demonstrates improved performance in cross-user hand pose estimation, particularly in settings with higher channel densities (16 and 110 channels). The authors conducted several ablation studies, which further identified high channel density as a significant factor for the effective pretraining of the proposed method.

### **Strength**
The experiments into cross-channel experiments, particularly concerning channel density and device configuration, addresses a relevant and significant research question within the community.

### **Weakness**
A primary concern of the paper is the limited scope of the experimental results.
1. The paper could benefit from additional inter-subject or inter-session variability results. It would be beneficial to analyze the performance per subject and per session, quantify the variation observed, and discuss potential explanations for such variability. The only results presented in the paper is just a summation of all the variabilities.
2. Evaluation in non-cross-subject settings. The authors could consider evaluating the model's effectiveness when not performing cross-subject evaluation. Would the performance significantly increase?
3. The authors can perform more comprehensive experiments to elaborate on how channel-density specifically influences model performance. This aspect of the results is not extensively explored and warrants further investigation to determine necessary conditions.
- First, the authors should do more data analysis and experiments to clarify whether observed effects are solely due to channel density, or if dataset differences or subject differences contribute.
- Given that the experiments are based on 110 channel data, the authors can systematically investigate downsampling 110 channels to 64, 32, 16, and 8 channels, and performing the exact same pretraining (instead of only downsampling it to 8 channels). Does the performance differ? Would any observed differences be attributable to channel position (e.g., varying downsampling strategies from 110 channels to 32 channels by considering different muscle groupings or different strides)?
4. The authors should perform analysis of model performance across different hand poses (targets). Specifically, determine if the best-performing class differs between high and low-density channel configurations.
5. Clarity on masking strategy. The explanation of the cross-block masking strategy could be enhanced. Providing a formal mathematical formulation of the masking strategies in the methodology, in addition to visualizations, would be beneficial.
6. Lack of benchmark methods. It would strengthen the paper to compare the proposed method with other existing methods and illustrate their performance differences.

Other concerns includes:
1. The information leakage argument appears questionable. A convolutional layer, when designed with appropriate kernel and stride sizes, could exhibit similar "information leakage" characteristics to an MLP layer employing a time-series patchify design with specific patch lengths and overlaps.
2. Overall, the observed performance improvement appears incremental. In Table 5, random initialization already yields performance values of 8.37 and 8.41, which are highly comparable to the random token masking value of 8.38.
3. What is the standard deviation of the network's performance?

**Audience:**

Yes

**Audience Explanation:**

The paper does perform investigations into different sEMG channel density and configuration, which addresses a relevant and significant research question within the community. While the experiments are not well designed or comprehensively performed, the problem setting is interesting, and will attract some TMLR's audience.

**Claims And Evidence:**

No

**Claims Explanation:**

Given the limited scope of experimental results, it is challenging for me to be convinced that the claims are well supported or accurate.

**Requested Changes:**

I'd request changes on all weaknesses of experimental results that are listed above.
1. Inter-subject or inter-session variability checks, which should include both numerical evaluations and visualizations.
2. Experimental evaluation in non-cross-subject settings.
3. Elaborate on how channel-density specifically influences model performance, which should include (1) More downsampling numbers from the original 110 channel dataset; (2) Different downsampling strategies (or to say, different selections of the channels); (3) Corresponding visualizations.
4. Analysis of performance across different hand poses.
5. Clarity on masking strategy.
6. Implementation of at least 3 different benchmark methods, which should include (1) Some basic feature extraction ML methods, like hand-crafted features and a remapping to the hand pose; (2) MAE-like reconstruction based methods; (3) Other contrastive learning methods. Also, the authors should consider switching the transformer blocks up to convolutional architectures to see if the pretraining strategy itself is helpful, or is it dependent on one specific type of architecture.
7. Include standard deviation of the network's performance for all tables. Run the model across different random seeds.

---

### Review · Reviewer_R5J4 · 2026-03-02

**Summary Of Contributions:**

The paper proposes EMG-JEPA, a self-supervised architecture to learn representations from unlabeled sEMG data, alleviating the data scarcity problem in sEMG-based hand pose estimation. the paper extensively evaluates the proposed method across devices with varying channel densities (8, 16, and 110 channels). The results not only demonstrate substantial reductions in joint angle estimation errors but also reveal a critical channel-density threshold (≈16 channels), providing valuable insights for the scalable design of future wearable interfaces.

# Strength:
1. The adaptation of the Joint Embedding Predictive Architecture (JEPA) to sEMG data is a highly elegant and well-motivated design choice. Unlike traditional reconstruction-based self-supervised methods (e.g., Masked Autoencoders) that force the model to reconstruct noisy, raw sensor signals, EMG-JEPA predicts representations in the latent space. This architectural design naturally filters out low-level sensor noise and encourages the model to focus on high-level, semantic muscle activation patterns.

2. For downstream applications, the model shows robustness to inter-subject variability. Because the JEPA objective forces the network to capture high-level semantic invariants of muscle activation rather than raw signal artifacts, the downstream pose estimator naturally generalizes better to unseen users without requiring tedious user-specific recalibration.

# Weakness:
While the paper effectively demonstrates the superiority of EMG-JEPA over traditional supervised baselines, it lacks of comparison with other state-of-the-art self-supervised learning (SSL) paradigm for time-series (for example Masked Autoencoders, i.e. MAE). Without these baselines, it is difficult to fully attribute the performance gains specifically to the predictive nature of JEPA rather than just the general benefit of using unlabeled data.

**Audience:**

Yes

**Audience Explanation:**

Yes. The findings of this paper will be of interest to several sub-communities within TMLR's broad audience. For example, researchers working on continuous hand pose estimation and gesture recognition are constantly bottlenecked by the high cost of synchronous MoCap data. The proposed scalable framework directly addresses this pain point, offering a practical solution for real-world deployment. Specifically for researchers in Representation Learning and SSL, this work demonstrates how the JEPA architecture can be successfully applied to sEMG data.

**Broader Impact Concerns:**

No major concerns. Paper have explicitly acknowledged the critical issues associated with processing multi-channel sEMG signals, including data privacy, user consent, and the necessity of demographic fairness.

**Claims And Evidence:**

No

**Claims Explanation:**

The authors provide a very convincing theoretical argument for choosing a predictive architecture (JEPA) over generative / reconstruction-based SSL methods, such as Masked Autoencoders (MAE), correctly pointing out that reconstructing highly noisy and stochastic raw sEMG signals is sub-optimal. However, the paper lacks an empirical comparison to validate this claim. Including a baseline comparison against a reconstruction-based MAE adapted for sEMG would significantly strengthen the argument that the performance gains stem specifically from JEPA's latent-space predictive nature, rather than just the general benefit of using unlabeled data.

**Requested Changes:**

Include a comparative experiment against a reconstruction-based SSL baseline (such as a MAE adapted for sEMG). This is critical to substantiate the claim and to demonstrate that the performance improvements are specifically due to the latent-space predictive objective of JEPA, rather than merely the general benefit of using large-scale unlabeled data for pre-training.

---

### Review · Reviewer_42De · 2026-03-09

**Summary Of Contributions:**

To address the limitations of existing hand pose estimation approaches—such as inconvenient wearable devices that may affect measurement accuracy and the difficulty of obtaining large-scale annotated datasets—the paper proposes EMG-JEPA, a self-supervised framework based on the Joint Embedding Predictive Architecture (JEPA). The proposed method applies spatiotemporal masking to sEMG signals and learns to predict the latent representations of the masked tokens from the visible context. Through this self-supervised learning paradigm, the model can capture meaningful spatiotemporal structures in sEMG signals without relying on explicit annotations. As a result, the approach reduces the dependence on large labeled datasets and has the potential to improve cross-user generalization.

**Audience:**

Yes

**Audience Explanation:**

The findings of this paper may be of interest to researchers working on sEMG signal modeling, hand pose estimation, and related human–machine interaction applications. The proposed EMG-JEPA framework addresses a common challenge in this domain, namely the scarcity of large-scale labeled datasets. By introducing a spatiotemporal masking mechanism, the model is able to learn the underlying spatial and temporal structures of sEMG signals (e.g., cross-electrode and temporal dependencies) in a self-supervised manner. This design enables the model to learn transferable representations without relying heavily on annotated data and may improve generalization across users. Therefore, the work could be relevant to researchers interested in biosignal representation learning, self-supervised learning, and practical sEMG-based applications.

**Claims And Evidence:**

Yes

**Claims Explanation:**

Key Strengths

- The figures in the paper are clear and informative. In particular, the illustrations in the Preparation of sEMG Signals for EMG-JEPA section provide an intuitive understanding of the sEMG signal acquisition and preprocessing pipeline, even without carefully reading the accompanying text.
- The proposed EMG-JEPA framework leverages self-supervised learning to reduce reliance on large-scale labeled datasets while improving the model’s ability to generalize across different users.

Key Weaknesses

- The Related Work section mainly lists previous approaches but lacks a clear discussion of their limitations. As a result, the motivation for the proposed method could be better articulated.

- In Section 3.2.2 (Architecture), the second paragraph includes several implementation details such as module counts and dimensional settings. These details would be more appropriately presented in the Training Details or implementation section to improve the overall organization of the paper.

- The technical novelty of the proposed framework may be limited, as it largely adapts the JEPA paradigm to sEMG signals. It would be helpful to clarify the methodological differences and advantages compared to existing self-supervised learning approaches such as MAE or contrastive learning frameworks.

- The idea of this paper is directly inspired by MAE, applying a masking strategy for self-supervised learning on sEMG data. Compared with other modalities, sEMG data may exhibit different characteristics and may require specific design choices. These aspects should be clearly explained.

**Requested Changes:**

- In the Related Work section, in addition to describing prior studies, it would be helpful to provide a clearer discussion of the limitations of existing approaches. Summarizing these limitations could better motivate the proposed method and make the transition to the authors’ contributions more natural. (Suggestion for improvement)
- In Section 3.2.2, the second paragraph contains several implementation details regarding the configuration and parameters of the encoder modules. These details might be more appropriately placed in the Appendix or an implementation details section, which would improve the clarity and organization of the main text. (Suggestion for improvement)
- In the Method section, the paper could more clearly highlight the methodological differences between the proposed approach and existing self-supervised learning frameworks (e.g., MAE or contrastive learning methods). Clarifying these distinctions and emphasizing the advantages of the proposed approach would strengthen the presentation of the work. (More important revision)

---

### Review · Reviewer_U8r1 · 2026-03-09

**Summary Of Contributions:**

### Summary:
The paper proposes a EMG-JEPA, a self-supervised learning framework for classification of EMG data. The method uses JEPA-based pretraining for learning features from unlabelled data, followed by finetuning on the target dataset. The key idea is to predict the latent representation from the visible context. Through multiple experiments and ablations, the paper demonstrates the usefulness of the method, consistently beating the baseline SFT method.

### Strengths:
The paper comes at the right time when there is a big conversation happening around the future of representation learning in ML. The paper's contribution could go well beyond just EMG, and highlight JEPA's validity for timeseries in general. Engineering contributions like tokenization, context/target encoders, spatiotemporal latent representation masking could translate to other domains.

### Weaknesses:
While the paper is promising, it lacks experiments to validate the claims. My major concern is that the results are presented (presumably) for only one split of train-test. Since the experiments are not repeated across multiple splits, it is hard to attribute the performance gains to just JEPA. In addition, there are no baselines on other self-supervised learning methods such as those based on masked autoencoders.

**Audience:**

Yes

**Audience Explanation:**

The paper is a timely contribution to the field of self-supervised learning, and readers would benefit from seeing how JEPA-like methods actually work in practice on timeseries.

**Claims And Evidence:**

No

**Claims Explanation:**

- Lacking enough experiments
- Results are not averaged over multiple splits
- Lack of strong baselines

**Requested Changes:**

- Repeated experiments across multiple splits. Present both the means and standard deviation.
- Include baselines based on a) masked-autoencoders, and b) lightweight networks with SFT that can work well in low-data regime.

---

### Author Response · Authors · 2026-03-11
**Withdrawal of Manuscript Submission**

Dear Xiaofeng and Reviewers,

I hope this message finds you well. I would like to express my sincere gratitude for the time and effort you invested in reviewing our manuscript, “EMG-JEPA: Towards Scalable and Generalizable sEMG-Based Hand Pose Estimation via Self-Supervised Learning”, and for providing valuable feedback.

Due to our company's internal review policies, we regret to inform you that we must withdraw our submission at this time. We appreciate the insightful comments and suggestions from both the editor and the reviewers, which will be instrumental in improving our paper draft.

We intend to resolve these internal matters promptly and plan to resubmit our revised manuscript once the issues are addressed.

Thank you once again for your understanding and support.

Sincerely,
Ali Nabi

---

### Note · Authors · 2026-03-11

**Comment:**

Due to our company's internal review policies, we regret to inform you that we must withdraw our submission at this time. We appreciate the insightful comments and suggestions from both the editor and the reviewers, which will be instrumental in improving our paper draft. We intend to resolve these internal matters promptly and plan to resubmit our revised manuscript once the issues are addressed.

**Withdrawal Confirmation:**

I have read and agree with the venue's withdrawal policy on behalf of myself and my co-authors.